# Asymmetric Gait Analysis Using a DTW Algorithm with Combined Gyroscope and Pressure Sensor

**DOI:** 10.3390/s21113750

**Published:** 2021-05-28

**Authors:** Yeon-Keun Jeong, Kwang-Ryul Baek

**Affiliations:** School of Electronics Engineering, Pusan National University, Busan 46241, Korea; busker0226@pusan.ac.kr

**Keywords:** gait analysis, asymmetry gait, inertial sensor, smart shoes

## Abstract

Walking is one of the most basic human activities. Various diseases may be caused by abnormal walking, and abnormal walking is mostly caused by disease. There are various characteristics of abnormal walking, but in general, it can be judged as asymmetric walking. Generally, spatiotemporal parameters can be used to determine asymmetric walking. The spatiotemporal parameter has the disadvantage that it does not consider the influence of the diversity of patterns and the walking speed. Therefore, in this paper, we propose a method to analyze asymmetric walking using Dynamic Time Warping (DTW) distance, a time series analysis method. The DTW distance was obtained by combining gyroscope data and pressure data. The experiment was carried out by performing symmetrical walking and asymmetrical walking, and asymmetric walking was performed as a simulation of hemiplegic walking by fixing one ankle using an auxiliary device. The proposed method was compared with the existing asymmetric gait analysis method. As a result of the experiment, a *p*-value lower than 0.05 was obtained, which proved that there was a statistically significant difference.

## 1. Introduction

Walking is a basic activity for all human beings and is an intuitive element for assessing health and quality of life. A normal gait leads to a healthy life, but an abnormal gait can worsen health. Conversely, an abnormal gait can also occur due to various diseases. By monitoring and evaluating gait, physicians can identify health conditions and predict various diseases. Gait as an indicator can help not only with the prediction of a disease but also with the degree of improvement in disease recovery or physical rehabilitation [1]. In hospital practice through the 6-Minutes Walking Test (6MWT), a doctor directly observes the number of steps, the walking time, and distance to determine disease progression or the degree of rehabilitation [2]. Among the methods for evaluating an abnormal gait, a method for determining asymmetry in gait is typically used [3].

Muscle strength asymmetry is the typical cause of asymmetric gait [4]. Stroke is one of the many causes of muscle strength asymmetry. In the field of gait analysis, the study of post-stroke hemiplegic gait occupies a large part after Parkinson’s disease, cerebral palsy, and orthoses [1]. Studies on various environments such as analysis of post-stroke hemiplegic gait according to the type of shoes [5] and repeated experiments on stroke patients [6] were also conducted. Timed Up and Go (TUG) test is being used as a method to determine the degree of disease progression or recovery for stroke patients [7]. In the past, the experiment on walking was observed and measured by a doctor. However, the recent test is a digital automation system using various sensors to collect and analyze data on walking [8,9,10]. Representative methods for determining asymmetric gait to analyze asymmetric gait after stroke include a discrete approach, a full gait cycle approach, a statistical approach, and a nonlinear approach [11]. Usually, to apply most of the methods listed above, the gait test measures the spatiotemporal parameter [12]. Typical spatiotemporal parameters are step length [13,14], step time [15,16], swing period [17,18], and step speed [19], and the degree of asymmetry is determined using the exponent between the parameters. However, the spatiotemporal parameter has the disadvantage of not considering the influence of the pattern diversity [20] and the walking speed [21]. On the other hand, few studies have been conducted using pattern similarity analysis of time series data [22]. Time-series data analysis is the analysis of specific patterns or sequences, also known as sequence patterns or signal patterns. This pattern analysis is useful for identifying differences in similar movements. This method of signal pattern analysis can intuitively grasp the overall trend, but the objectivity of the study may be impaired. Therefore, in this paper, we propose a method to analyze gait asymmetry that guarantees objectivity by applying a Dynamic Time Warping (DTW) algorithm. The DTW algorithm requires more calculations than conventional descriptors that use common spatial parameters such as stride time, swing time, etc. The methods that use spatial parameters are relatively simple because only the discrete events of the gait cycle are analyzed. Contrarily, the DTW algorithm requires more calculation power because it uses continuous signals for gait analysis. As a result, a more precise asymmetric gait analysis is obtained.

In applying the DTW algorithm, gait data were collected through gyroscope and pressure sensors. There have been many studies on asymmetric gait using gyroscopes [23,24] and pressure sensors [25]. However, the data of the asymmetric gait analysis using the DTW algorithm for the data combining the two sensors was not found. Therefore, in this paper, we propose a method of combining time-series data from gyroscope and pressure sensors. The DTW algorithm is applied to the combined data, and the asymmetry of the gait is evaluated and analyzed.

## 2. Materials and Methods

As shown in Figure 1, the gait cycle is divided into the stance and the swing. The stance phase is from the beginning of the heel strike (HS) to the toe-off (TO), and the swing phase is from the TO to the HS. In the gait cycle, the stance phase accounts for 60% of the total and the swing phase accounts for 40% [26]. The stride is when the stance and swing occur once. That is, one stride can be seen as the interval between each HS. Also, the mid-stance (MS) is the point where the foot flatly touches the ground, and the angular velocity is zero; it occurs between the HS and the TO in the stance phase. The mid-swing (MSW) is between the TO and HS of the swing phase, where the angular velocity switches from acceleration to deceleration.

The hardware configuration is shown in Figure 2. When the PC sends a trigger signal, the IMU module and the pressure sensor-equipped Arduino each transmit data to the PC. The IMU module transmits data according to the manufacturer’s proprietary protocol based on the trigger signal. The Arduino platform converts the analog signal from the pressure sensors into a digital signal based on the trigger signal and transmits the data using a protocol that is directly configured.

### 2.1. Sensor Installation and Data Acquisition

As shown in Figure 3, the IMU sensor was attached to the tip of the shoe and enclosed in a case made with a 3D printer. Also, axis information is displayed in the right-hand image of the figure. Only the angular velocity of the *x*-axis was obtained from the installed inertial sensor.

Figure 4a shows the pressure sensor attached to the insole at the toes and the heels. In general, a pressure sensor collects data from each of the sensors [27,28,29], but in this paper, we propose a method of combining two pressure sensors to obtain the data. The pressure sensor used is a force-sensing resistor (FSR) sensor, and the resistance value changes depending on the pressure. Therefore, as shown in Figure 4b, two sensors (resistors) were configured as voltage dividers. The output value for the voltage-divided sensor is the balance value for the forefoot and heel. In this paper, since time series data are analyzed, accurate pressure values are meaningless. By configuring two sensors into one output, data can be acquired with one ADC channel. In this way, the hardware cost can be reduced even if it is a small amount.

Figure 5 shows the walking pressure-sensor output for the proposed installation method. The negative values indicate when the heel of the foot touches the floor, and the positive values are when the forefoot touches the floor. The zero sections are when the foot is completely off the ground. Actually, V_in_ of the voltage divider in Figure 4b is 5 V, and when the pressure of the two sensors is 0, V_out_ takes 2.5 V, which is half of 5 V, but the offset of 2.5 V is removed after data acquisition.

### 2.2. Signal Processing for Asymmetric Analysis

The obtained gyroscope and pressure data were put through the process shown in Figure 6 to analyze the asymmetry. The rule-based event detection is performed (Section 2.2.1). Based on the detected events, strides are extracted from the gyroscope and pressure data and collected, and then the two data sets are superimposed (Section 2.2.2). Finally, the DTW distance is calculated for time series analysis of gait asymmetry (Section 2.2.3).

#### 2.2.1. Rule-Based Event Detection

The first task is to distinguish the stride in the data obtained from the gyro sensor. The stride is detected from the filtered gyro data through a rule-based event detection process. Event detection is a preprocessing process for classifying strides before determining gait asymmetry, and it is necessary to detect it online to build an automated system. Therefore, in this paper, we propose a rule-based event detection method. Figure 7 shows the result of gait event detection.

As mentioned in Figure 1, the stride is between heel-strike. Therefore, the final aim is to detect each HS. In this paper, after detecting the MSW (red triangle) and the MS (green triangle), each HS (black triangle) is detected as the minimum value between the two points. The MSW is detected as the maximum value between the points of increasing (yellow triangle) and decreasing (magenta triangle) value based on a specific threshold (the dotted red line). The threshold was set to five degrees per second. The MS is detected as a point at which the difference between the previous value and the current value is smaller than a specific threshold, within a range based on 0 points. The range (the dotted blue line) is ±1.5°/s, and the threshold is 0.25°/s. In addition, TO (cyan triangle) is also detected to compare with the existing spatiotemporal parameters (stride time, swing time). TO is simply detected as the minimum value that exists in the range between MS and MSW. To verify the reliability of the proposed rule-based event detection method, 459 steps were recorded by 5 adult males. The successful detection rate was 98.257% (451 steps were accurately detected out of 459). In this paper, a detection rate of 98.257% is an appropriate value because the mean stride is obtained after each stride is extracted to analyze asymmetrical steps as shown in Figure 7.

#### 2.2.2. Stride Extraction and Resampling and Data Fusion

The stride segments extracted through rule-based gait event detection are resampled to a gait cycle ranging from 0–100% on the *x*-axis. The resampled segments are aggregated and averaged. The resampling is implemented with a function provided by MATLAB using the Piecewise Cubic Hermite Interpolating Polynomial (PCHIP) algorithm. In addition, standardization is performed on the *y*-axis to combine the gyroscope and pressure data. Resampling (or normalizing) in the same way as the *x*-axis results in loss of information about the amplitude difference between the gyroscope data and the pressure data for symmetric or asymmetric gait. Therefore, the gyroscope and pressure data are standardized as shown in Equation (1). The same resampling process applies to pressure data as well as gyroscope data.
(1)Z=(X−μ)/σ

In this paper, to analyze gait asymmetry, time-series data is compared, not a numerical comparison. Therefore, the standardized gyroscope and pressure data ware fused using the principle of superposition for time series data analysis. The data extracted for the same stride have the same time-series data. Therefore, the two sets of data are regarded as a waveform for the same time and were fused using a waveform overlap.

#### 2.2.3. Dynamic Time Warping

Dynamic Time Warping is a time series analysis algorithm that measures the similarity between two sequences of different speeds. Therefore, voice recognition using the DTW algorithm is a very active field of research [30]. Generally speaking, the speed of speech varies from person to person, resulting in nonlinear fluctuations in the speech pattern versus the time axis. To eliminate this, the DTW algorithm is applied to achieve different temporal coincidences through time warping. The speed of the gait is different for each person, and the swing of the left foot and right foot and the reaction force on the ground is different creating asymmetry. Therefore, the DTW algorithm can also be applied to the asymmetric analysis of the gait. In gait analysis, the DTW algorithm is mainly used for gait classification [31], clustering [32], and stride segmentation [33]. On the other hand, there are not many studies on asymmetric gait analysis applying the DTW algorithm. In this paper, we analyzed asymmetry by calculating the time warping of time series data from the left and right feet using the DTW algorithm.

The DTW algorithm has two steps. The first is to create a cost matrix. A sample sequence for the left foot and the right foot can be expressed as follows:(2)$$X=\left[x_{1},x_{2},\;\ldots ,\;x_{m}\right]\;\;\;Y=\left[y_{1},y_{2},\;\ldots ,\;y_{n}\right]$$

Assuming that the number of samples in each sequence is m and n, cost matrix *D* becomes m×n. The elements of the cost matrix are created from the following equation.
(3)$$D\left(i,j\right)=Dist\left(i,j\right)+min\left\{\begin{array}{c} D\left(i-1,j\right)\\ D\left(i,j-1\right)\\ D\left(i-1,j-1\right) \end{array}\right\}\;Dist\left(i,j\right)={\left(x_{i}-y_{j}\right)}^{2}$$

Each element is calculated as the sum of the distance of the index (absolute value of the difference between the sample values) and the minimum value among neighboring elements.

The second step is to construct the warp path W using the cost matrix *D*. The warping path W from (1,1) to (n,m) is represented by Equation (4), and each element, $w_{k}={\left(i,j\right)}_{k}$, is the Euclidean distance between $x_{i}$ and $y_{j}$. As shown in Equation (5), DTW is the lowest cost path for all possible paths. Consequently, the DTW distance is expressed by Equation (6).
(4)$$W=\left[w_{1},w_{2},\ldots ,w_{k}\right]$$
(5)$$DTW=min\overset{k}{\underset{x=1}{\sum }}W_{x}$$
(6)DTW(X,Y)=D(n,m)

### 2.3. Subjects and Experiment Protocol

The gait experiment was conducted with 5 members (five males) of the laboratory. The members are physically healthy and participated in the experiment with a sufficient understanding of the purpose of the experiment. Their average age is 27.2 years, the average height is 1.77 m, and the average weight is 72.6 kg. For each member, 10 normal walks and 10 asymmetric walks were performed. This study was approved by the Bioethics Committee of Pusan National University, the author’s affiliation, and each subject signed an informed consent form. The walking experiment was carried out in a straight corridor of about 40 m, as shown in Figure 8.

To determine gait asymmetry, tests were performed with normal and asymmetry gait. Asymmetry gait was performed by simulated gait for hemiplegic patients. The gait of hemiplegic patients is asymmetric because of a paralyzed ankle. Therefore, for a more complete simulation of walking, paralysis symptoms were simulated by wearing an orthosis on one ankle as shown in Figure 9.

## 3. Results and Discussion

Figure 10 shows the extraction results and average values of the gyroscope and pressure data for the left and right feet. Figure 10 shows the extraction results and average values of the gyroscope and pressure data for the left and right feet. The data is the walking data of Subject 1 and is the result of one walk (about 30 m straight symmetrical walking).

Figure 11 shows the combined data of standardized gyroscope and pressure data for symmetric and asymmetric walking. The left side of the figure is the result of symmetrical walking, and the right side is the result of asymmetrical walking. The black line is data for the left foot, and the red line is data for the right foot.

Also, the DTW result graph and DTW distance for the combined data are shown in Figure 12. As the DTW distance becomes smaller, the two sequences can be regarded as similar, and the DTW distance of two completely identical sequences has a value of 0.

Table 1 shows the results of the existing method and the proposed method for symmetric and asymmetric gait as mean, standard deviation, and *p*-value. To test the statistical difference, the *p*-value was performed with the Mann–Whitney test. The proposed method was compared with the existing methods such as stride duration, swing duration [4], and maximum angular velocity [34]. Stride duration and swing duration mean the time for each gait phase, and the maximum angular velocity is the maximum value of the gyroscope data. Existing methods are symmetry indexes, that is, a relative parameter for the data of both feet. Therefore, the result of all methods is zero when walking completely symmetrically. DTW distance is a parameter for time series data combined with gyroscope and pressure data, which is a method proposed in this paper.

As shown in Table 1, in all methods except stride duration, the *p*-value is less than 0.05. For hemiplegic gait, the stride duration indicates that there is no significant difference. In addition, the mean and standard deviation of the other methods are significantly different in the results of asymmetric gait compared to symmetric gait. Stride duration is the time interval between HS, so there is no difference in hemiplegic gait. However, due to hemiplegia, the swing duration increases and takes up a large proportion compared to the symmetric gait. Therefore, the swing duration ratio made up of the logarithmic function shows a large difference for negative values. In addition, hemiplegic gait has a limited rotation of the ankle and knee. Consequently, a large difference appears in the results of the maximum angular velocity. The proposed method, DTW distance, is also time-series data in which gyroscope data and pressure data are combined, and as a result, there is a large difference between symmetric and asymmetric gait.

Figure 13 shows the results for all subjects. As shown in Table 1, the asymmetry can be sufficiently confirmed in the proposed method including the existing method excluding the stride duration. Therefore, the result of the proposed method, the DTW distance, is considered as a criterion for determining the symmetry of the gait along with the result of the existing asymmetric gait analysis method.

Figure 14 shows an analysis of the relationship between gyroscope and pressure data. As shown in Figure 14, the pressure data time shift was observed for the gait event in each step. Figure 14a shows the gyroscope and pressure data observed in a normal gait. The data represent two strides. The graph shows that in the MS immediately after the HS when the gyroscope data value is close to 0, the pressure data value shows that the force applied to the ground moves from the heel (negative peak) to a positive peak. Next, the graph shows that from TO to HS (that is, in the swing section), the pressure data value is 0. When the HS event occurs, you can see that the gyroscope data value decreases from the positive peak when the pressure data value is negative. However, Figure 14b shows that before the gyroscope data value reaches a positive peak, the pressure data value starts to decrease from zero to a negative value. That is, in normal walking, when the gyroscope data value is a negative peak, it is regarded as HS. This means that the ankle is turned to the maximum in the HS event by the swing. Therefore, as shown in Figure 14a, when the gyroscope data value is a negative peak, the value of the pressure data value follows as a negative peak in the normal step. However, as shown in Figure 14b, the value of pressure data value reaches a negative peak before the value of the gyroscope data reaches a negative peak. This means that the foot touches the ground before the ankle is fully rotated. This result shows the asymmetry of hemiplegic gait with respect to the correlation between ankle rotation and foot contact with the ground. This characteristic of asymmetric gait can be a criterion for distinguishing another disease or can determine the degree of recovery from a disease [35,36]. In addition, the DTW distance, which is a combination of gyroscope data and pressure data, is expected to apply to asymmetric gaits, such as freezing of the gait (FOG), a foot drop gait (FD), and a slapping gait, which correlates with ankle rotation and foot contact with the ground.

## 4. Conclusions

The method proposed in this paper analyzes time series data that combines gyroscope data and pressure data and proposes parameters for asymmetric walking. Data was acquired by attaching a gyroscope sensor in front of the shoe and a pressure sensor on the insole. Gait data were acquired through normal symmetric walking and simulated asymmetric walking. The gyroscope and pressure data acquired through the gait experiment were combined by detecting gait events, extracting gait, resampling and standardizing, and superimposing. The combined data was compared with the existing asymmetric gait analysis method by calculating the DTW distance through the DTW algorithm. The process of obtaining the DTW distance using the acquired sensor data has been implemented online.

In previous studies, asymmetric gait was analyzed and compared with indexes for stride duration, swing duration, and maximum angular velocity. Due to the characteristics of hemiplegic gait, the stride duration was inadequate for asymmetric gait, but the swing duration, maximum angular velocity, and the proposed method showed satisfactory results. The *p*-value was less than 0.05, indicating the experimental results were statistically significant. Therefore, this paper provides a new indicator that can be used in the study of asymmetric gait.

Future research is a study to segment and analyzes the characteristics of asymmetric walking through numerical analysis as well as DTW distance for time series data combined with gyroscope data and pressure data. In addition, this research can be a tool to classify hemiplegic patients with machine learning techniques, i.e., the support vector machine (SVM) and k-nearest-neighbor (KNN). Finally, to use the proposed method in an asymmetric gait analysis system or a monitoring system that determines the degree of recovery, further experiments on patients with asymmetric gait or hemiplegic are required.

## Figures and Tables

**Figure 1 sensors-21-03750-f001:**
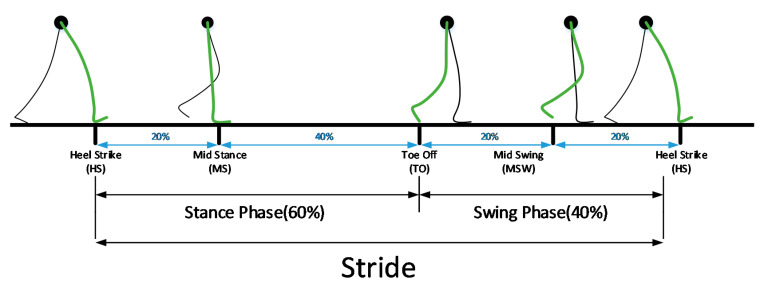
The stride (gait cycle) is divided into the stance phase and the swing phase.

**Figure 2 sensors-21-03750-f002:**
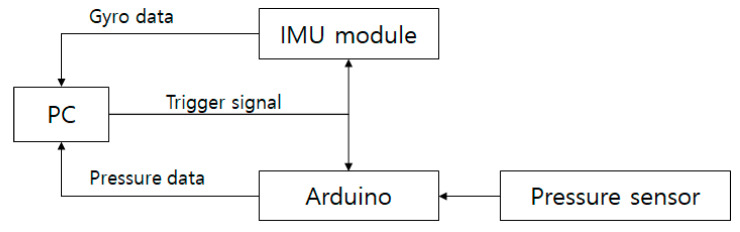
Hardware configuration.

**Figure 3 sensors-21-03750-f003:**
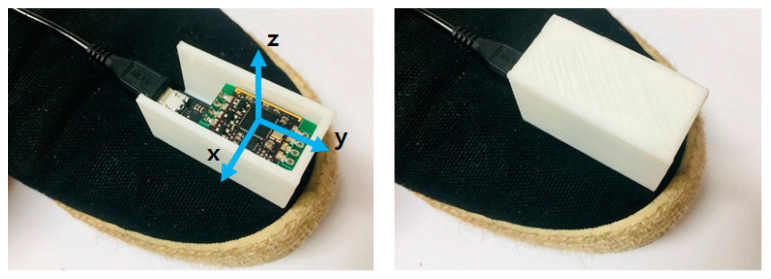
The inertial sensors, in a case made with a 3D printer, are attached to the shoe’s tip.

**Figure 4 sensors-21-03750-f004:**
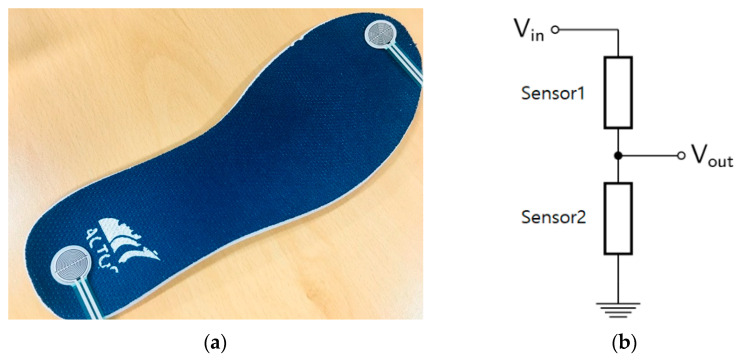
Pressure sensor installation and circuit configuration. (**a**) pressure sensors attached to the insole; and (**b**) voltage divider composed of pressure sensors.

**Figure 5 sensors-21-03750-f005:**
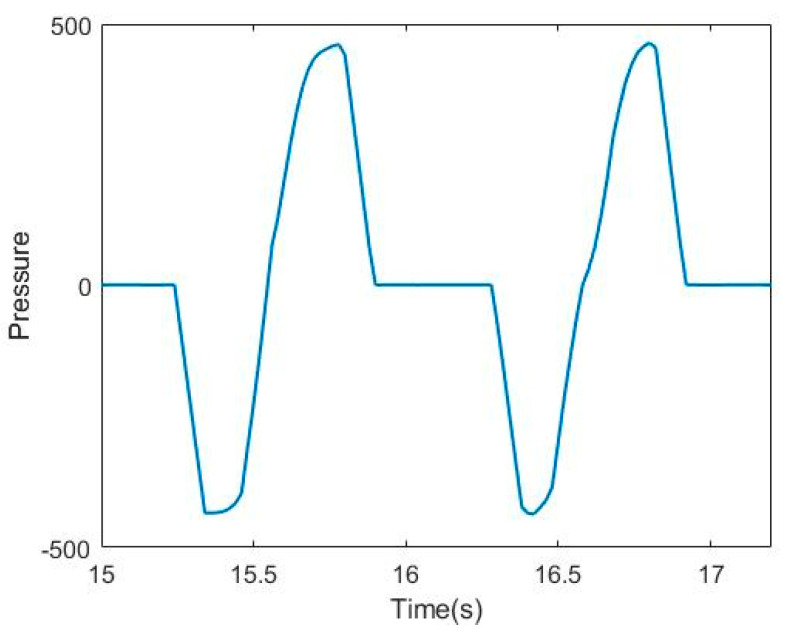
Pressure sensor data.

**Figure 6 sensors-21-03750-f006:**
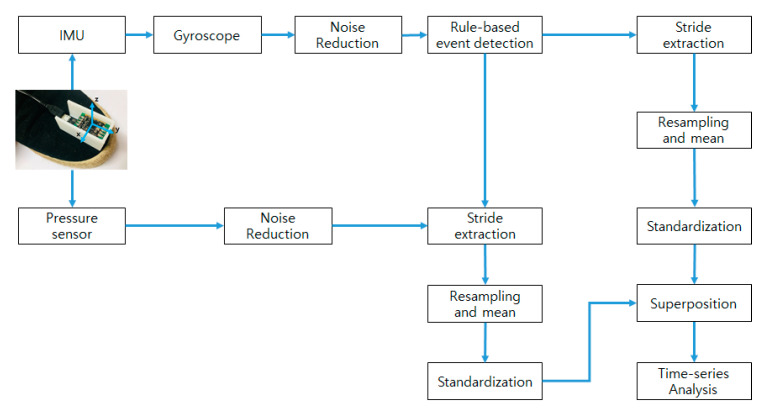
Flow chart for signal processing.

**Figure 7 sensors-21-03750-f007:**
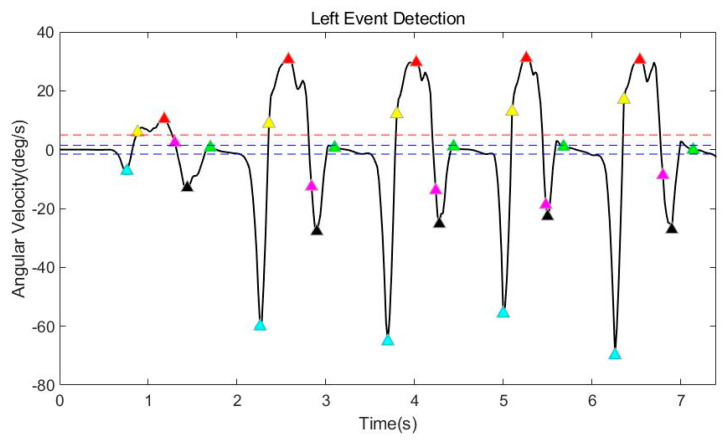
The result of gait event detection.

**Figure 8 sensors-21-03750-f008:**
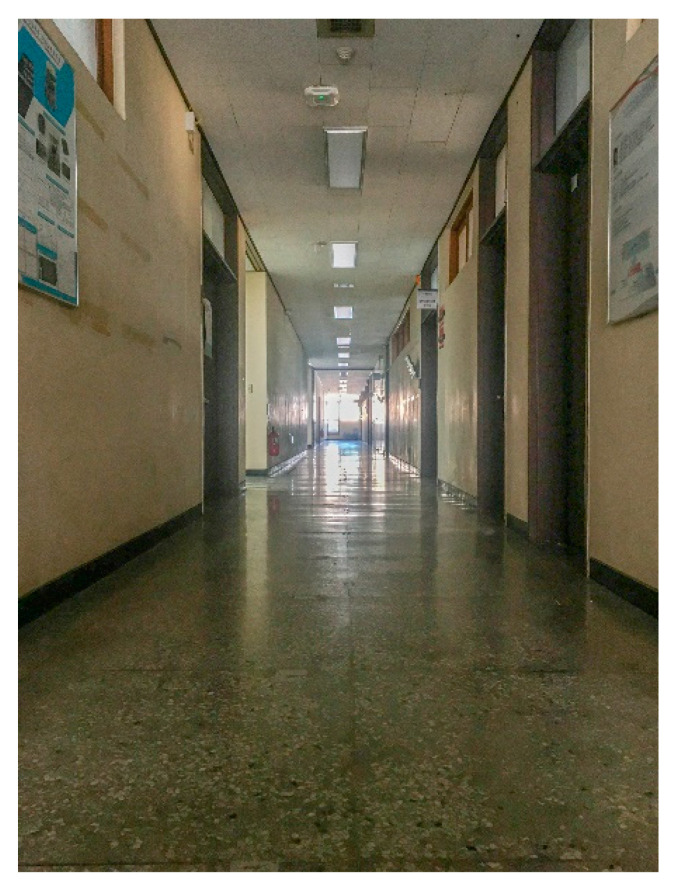
The environment for the walking experiment.

**Figure 9 sensors-21-03750-f009:**
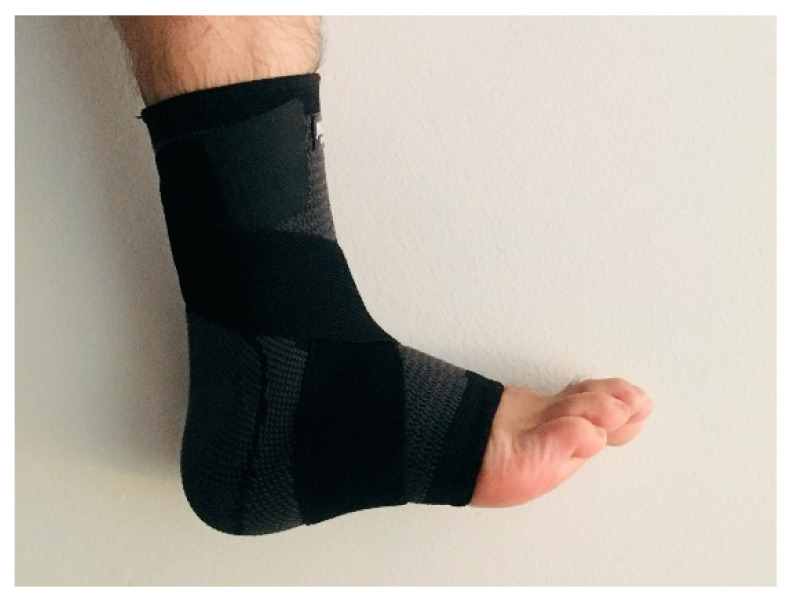
A useful orthosis for simulating ankle paralysis.

**Figure 10 sensors-21-03750-f010:**
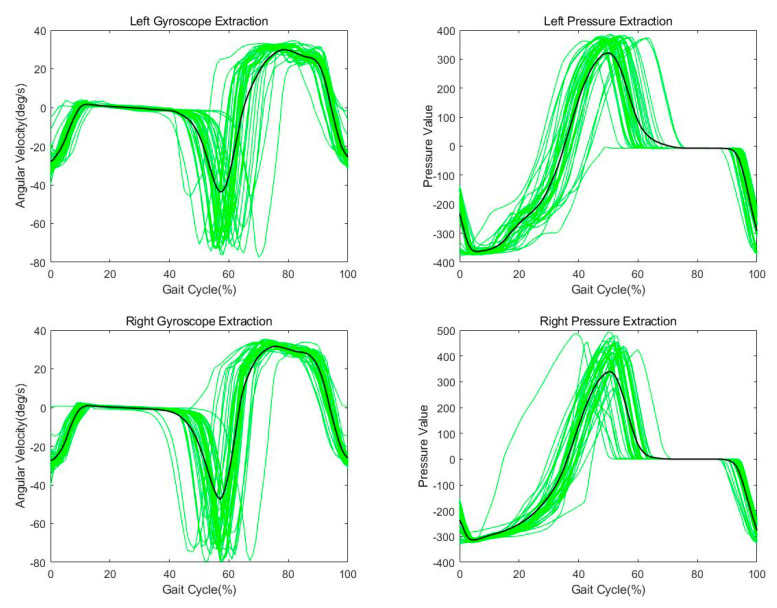
Stride extraction and mean from the gyroscope/pressure data.

**Figure 11 sensors-21-03750-f011:**
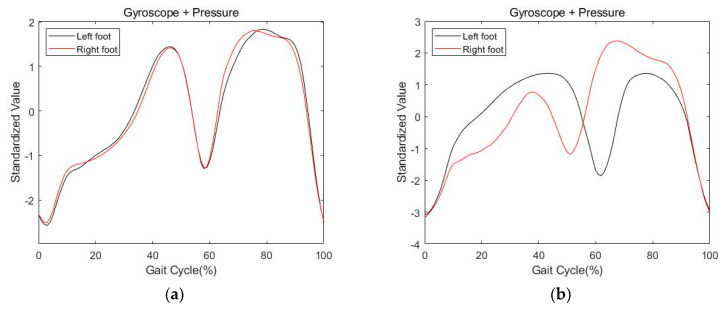
The results of combined data. (**a**) Symmetric gait; (**b**) asymmetric gait.

**Figure 12 sensors-21-03750-f012:**
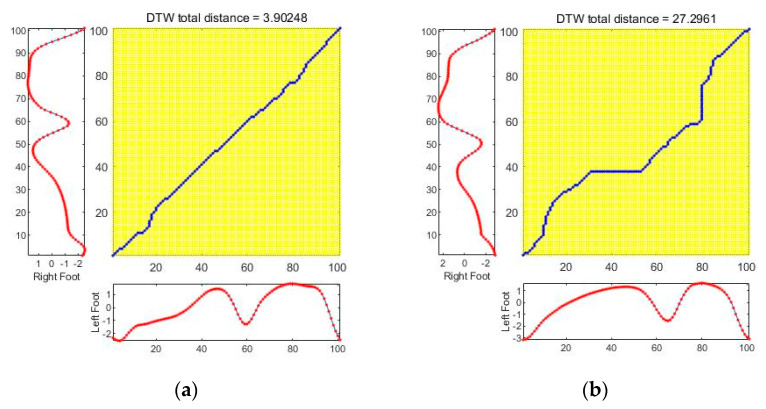
The results of DTW. (**a**) Symmetric gait; (**b**) asymmetric gait.

**Figure 13 sensors-21-03750-f013:**
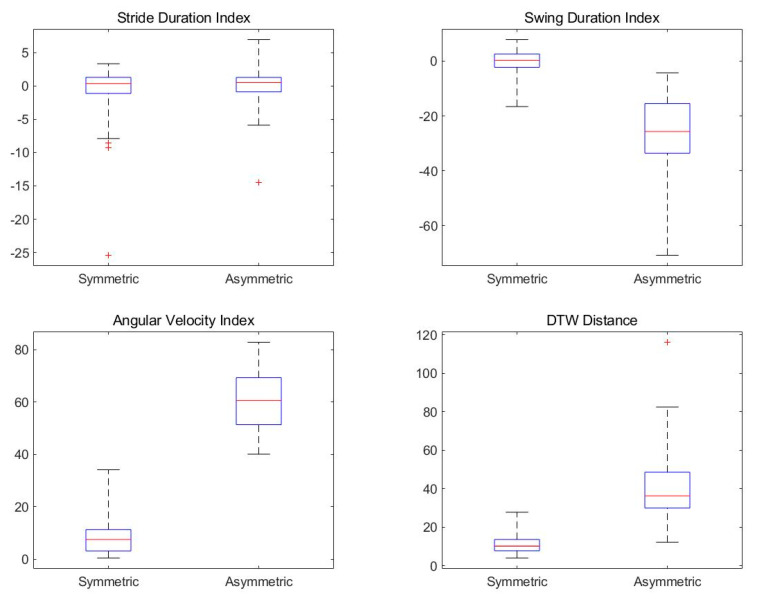
Comparison of results for all subjects.

**Figure 14 sensors-21-03750-f014:**
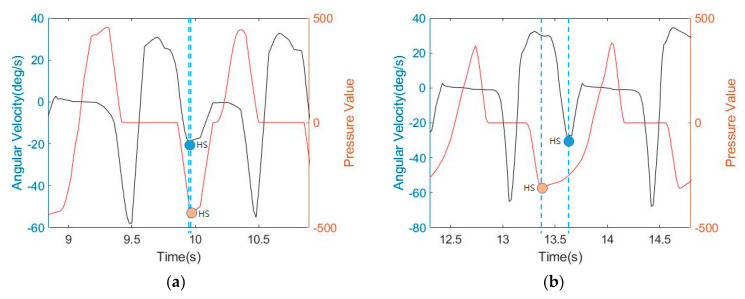
Comparison of gyroscope and pressure sensor data. (**a**) symmetric gait; and (**b**) asymmetric gait.

**Table 1 sensors-21-03750-t001:** Mean, standard deviation, and *p*-value for the analysis methods.

Subjects	Method	Symmetric Gait	Asymmetric Gait	*p*-Value
Subjects1	Stride duration	−0.99 ± 4.03	−0.56 ± 1.44	0.7937
	Swing duration	−1.05 ± 4.62	−33.38 ± 6.71	0.0002
	Maximum angular velocity	7.30 ± 6.84	52.32 ± 7.49	0.0001
	DTW distance (proposed)	13.34 ± 4.66	33.94 ± 5.25	0.0002
Subjects2	Stride duration	0.15 ± 1.98	2.16 ± 2.39	0.0643
	Swing duration	2.02 ± 4.87	−9.44 ± 4.26	0.0003
	Max angular velocity	8.90 ± 9.44	58.30 ± 8.64	0.0002
	DTW distance (proposed)	11.41 ± 5.38	32.44 ± 11.40	0.0004
Subjects3	Stride duration	0.90 ± 1.34	0.30 ± 0.98	0.8493
	Swing duration	0.04 ± 2.44	5.71 ± 4.52	0.0002
	Max angular velocity	5.71 ± 4.52	57.83 ± 8.82	0.0008
	DTW distance (proposed)	8.59 ± 2.18	46.72 ± 12.60	0.0002
Subjects4	Stride duration	−0.11 ± 1.57	0.41 ± 2.24	0.3472
	Swing duration	0.32 ± 3.68	−17.59 ± 6.07	0.0002
	Max angular velocity	8.71 ± 4.35	61.44 ± 15.59	0.0002
	DTW distance (proposed)	15.15 ± 6.60	22.22 ± 7.49	0.0455
Subjects5	Stride duration	−4.31 ± 8.28	−3.02 ± 7.34	0.6745
	Swing duration	−3.15 ± 7.58	−38.92 ± 17.95	0.0002
	Max angular velocity	14.31 ± 7.09	74.13 ± 8.48	0.0003
	DTW distance (proposed)	9.53 ± 4.53	72.48 ± 26.34	0.0002

## Data Availability

Not applicable.

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
