# Peer review of "Asymmetric Gait Analysis Using a DTW Algorithm with Combined Gyroscope and Pressure Sensor"

_sensors, 2021, doi:10.3390/s21113750_

Round 1
Reviewer 1 Report
The paper is much better introduced and structured. The changes made greatly improve readability and clarity. However, I am surprised that the authors did not process classification. The descriptor is given (DTW) and we have samples, and this is the final objective of the process? We would have more relevant criteria than the p-value (which might need to be presented a little).
The contribution of the approach should also be better highlighted. Knowing that the DTW require much more calculation than the other descriptors for a result that is at best comparable.
Minor points:
-
A lot of redundancy in the pharses: the DTW distance of the DTW algorithm
-
L43-44: The field repartition of the review is not explained so not very informative.
-
L54: why does the DTW guarantee objectivity?
-
L167: it is equation (1) and not (2)
-
l196 Dist (1,1) = Dist (1,1) certainly but it is not really of interest.
-
Equation (5): Wx is a path (a vector of dimension x). What does the sum of vectors of different dimensions represent?
-
Figure 10 includes the asymmetric gait sequences or not?
-
Figure 11 and 12 are examples on a single sequence and or an average result?
Reviewer 2 Report
The revised version of the paper has all the comments from the reviewer applied. Nonetheless, the graphs and figure throughout the manuscript have very poor quality especially figures 2, 4, 7, 8, 9, 10, 11, 13 and 14 to a point that some texts are not illegible and cannot be read.
The authors are encouraged to follow the journal's instruction on the minimum figure/photo quality required.
Author Response
Please see the attachment.

This manuscript is a resubmission of an earlier submission. The following is a list of the peer review reports and author responses from that submission.
Round 1
Reviewer 1 Report
The paper's premise is not novel, but interesting nonetheless.
The are few points that need to be addressed:
Major:
- The abstract and conclusion are very basic and they are more inline with an undergraduate level coursework than a scientific paper.
- The literature review is very general, basic and not detailed. You need to discuss about some of the previous works' results as well as their methodologies.
- In line 76, it was mentioned that: "Noise was removed by using a moving-average filter, and the window size of the filter was set to five samples.". You need to clarify/justify why you chose the filter in the first place and also what was the logic/reason behind choosing with Window size of 5 rather than other values? Moreover, from the graphs provided, the before-filter graph is pretty much clean and the SNR is already high. What lead to the use of this filter?
- Line 200, discussed about the experimental design, this need to be in the methodology part not the results and discussion.
- Talking about experimental design, there is no mention of the number of subjects including their health condition or age/gender.
- The paper emphasis is on the asymmetry of Gait yet there is only one graph (Figure 9) showing left and right foot in which it seems symmetrical at least from the graph as there is not a table showing the differences.
- There is no table showing the results of individual participants with P values. The authors only provided one value as a result. Is it the average between all the participants? How many were participating? What was the standard deviation?
- There is no comparison of the results coming from pressure pad and gyro to a gold standard to infer the accuracy and precision.
Minor:
- In line 23, there is an abbreviation that has not been defined.
- The paragraph starting at line 44, has a very simple explanation that does not need to be there. Again, I get the feeling that this level of explanation in this paper is in line with a low-tier coursework rather than a scientific journal paper. I am sure the audience of this paper will know fully what Mid-stances or Heel strikes are.
Reviewer 2 Report
This paper presents a pipeline where gyroscope and pressure sensor data are fused and entered in a DTW that have to classify normal to abnormal gait.
This paper is first disturbing in its structure. The abstract and the introduction are very short and therefore introduce very poorly the context, the positioning of the method with the state of the art, the purpose of the method and its main steps.
The method section introduces technical details (values of the variables, experimental protocol, etc) which would be better placed in the experiments section. The presentation of the method then lacks rigor (see below).
The experimental section is not precise enough (how many different people) and is only carried out on healthy people. Only a graphic representation shows an isolated case without knowing whether it is said to be representative by an expert (such as the thresholds in Figure 8). There is no comparison and at the end the method does not classify gait.
So I think the classification should be done and the paper structure overhauled before publication.
Minor comments :
-
the meaning of the colors is not specified in the curves which prevents their understanding
-
in section 2.2.1 nothing is specified concerning the blue triangles
-
The presentation of the DTW lacks rigor:
1- equation 2 we must first initialize (+ infinity) the edges (first row and column)
2- specify that W is a path from (1,1) to (m, n)
3- DTW is finally the value D (m, n) -
Table 1: and what gives pressure alone?
-
Figure 13: how is the “dragged gait” obtained? simulated by an actor?
-
section 2.2 should not contain only a figure without description and analysis
-
a title have not be alone at the bottom of the page
-
6MWT not defined